# Structural Error Patterns Matter: Towards More Structure-aware GNN Evaluation and Training

## Abstract

Graph Neural Networks (GNNs) are a specialized family of neural networks designed to handle graph-structured data, enabling the modeling of complex relationships within graphs. Despite significant algorithmic improvements, the issue of performance evaluation for GNNs has largely been overlooked in the literature. A crucial but underexplored aspect of GNN evaluation is understanding how errors are distributed across the graph structure, which we refer to as the "structural error pattern". To the best of our knowledge, this paper is among the first to highlight the importance of paying attention to these error patterns, which are essential not only for model selection—especially in spatial applications where localized or clustered errors can signal critical issues—but also for providing algorithmic insights into the model's performance. In this work, we introduce a novel mathematical framework that analyzes and differentiates evaluation metrics based on their sensitivity to structural error patterns. Through a thorough theoretical analysis, we identify the limitations of traditional metrics—such as accuracy and mean squared error—that fail to capture the complexity of these error distributions. To address these shortcomings, we propose a new evaluation metric explicitly designed to detect and quantify structural error patterns, offering deeper insights into GNN performance. Our extensive empirical experiments demonstrate that this metric enhances model selection and improves robustness. Furthermore, we show that it can be incorporated as a regularization method during training, leading to more reliable GNN predictions in real-world applications.

## 1 Introduction

Graph Neural Networks (GNNs) have emerged as powerful models for analyzing graph-structured data, thanks to their ability to capture complex relational dependencies inherent in graph topologies. This makes GNNs particularly effective for node-level prediction tasks, such as classification or regression, where the target prediction is associated with individual nodes and influenced by their structural context and interactions with neighboring entities. As a result, GNNs have achieved significant success across a wide range of applications, including traffic forecasting (Zhao et al., 2019a; Guo et al., 2019; Zhang et al., 2020; Jiang & Luo, 2022), urban planning (Li et al., 2022; Chen, 2020), environmental monitoring (Zhang et al., 2023; Li et al., 2024), social network analysis (Kipf & Welling, 2017), and sensor network analysis (Dong et al., 2023; Saadati et al., 2024). In these contexts, data naturally form graphs, where nodes represent the entities of interest, and edges capture relationships such as proximity or connectivity. By integrating local node features with structural relationships across the graph, GNNs can deliver accurate, context-aware predictions that reflect the underlying structural dynamics and dependencies.

**Existing Gaps.** Most research on GNNs has focused on algorithmic innovations, architectural improvements, and system optimizations, aiming to enhance computational efficiency, scalability, or predictive performance. However, comparatively little attention has been given to developing evaluation metrics specifically tailored for network applications Bechler-Speicher et al. (2025). Rigorous evaluation frameworks are essential for reliably assessing model performance and guiding model selection, which directly impacts practical deployment and real-world effectiveness (Shchur et al., 2018; Dwivedi et al., 2023). In particular, for network applications, appropriate metrics should

not only quantify predictive accuracy but also capture critical characteristics of error distributions, such as clustering or dispersion patterns. **Without structure-aware evaluation, practitioners lack insights into how prediction errors manifest across the graph, hindering their ability to diagnose, address, and prevent localized failures effectively. Moreover, as demonstrated in (Huang et al.) and in this paper, structural error patterns provide valuable insights for improving GNN frameworks.**

**Limitations of Existing Evaluation Metrics.** Currently, GNN model evaluation predominantly relies on traditional metrics—such as accuracy (ACC) for node classification or mean squared error (MSE) for node regression tasks. While these metrics are widely adopted, they evaluate predictions independently at each node, ignoring the structural context and inter-node dependencies that are inherent to graph datasets. As we demonstrate both theoretically and empirically in this paper, such metrics fail to distinguish between different structural error patterns, including clustered errors and errors that are randomly distributed across the graph. This limitation is particularly problematic for real-time and fault-tolerant network systems. For instance, in traffic monitoring, where the graph structure represents spatial connectivity, clustered prediction errors may indicate localized congestion or sensor malfunctions that require immediate attention, whereas randomly dispersed errors may simply reflect minor inaccuracies (Xu et al., 2024; Fathurrahman & Gautama, 2024; Moretti et al., 2025). We illustrate this limitation further through a quantitative example in Appendix D.1. Consequently, the inability of conventional metrics to detect these structural error patterns significantly restricts practitioners' capacity for timely identification and intervention in critical regions. **Thus, developing evaluation frameworks explicitly designed to capture and quantify structural error distributions in GNN predictions is both urgent and essential.**

**Contributions.** In this paper, we address the under-explored yet crucial aspect of evaluating GNNs within network applications. Specifically, we examine widely used evaluation metrics and identify their inadequacies in capturing structural prediction error patterns. To overcome these limitations, we propose a novel structure-aware evaluation metric explicitly designed to quantify and differentiate structural error patterns, enabling precise detection of structural clustering in prediction errors. The key contributions and findings of this paper are summarized as follows:

• We develop a comprehensive mathematical framework to analyze evaluation metrics commonly employed in GNN tasks. Extending the concept of *expressiveness* (Definition 1), originally introduced in the context of the graph isomorphism problem, we formally define and analyze the property of *exchangeability* (Definition 2) inherent in traditional metrics such as ACC and MSE. Through rigorous theoretical analysis, we demonstrate critical limitations of exchangeable metrics, particularly their inability to differentiate distinct structural error patterns, such as clustered, dispersed, or randomly distributed errors (Theorem 3.1). This fundamental shortcoming emphasizes the necessity for metrics explicitly tailored to capturing structural error distributions in GNN predictions.

• Motivated by our theoretical insights, we propose a novel structure-aware evaluation metric, termed **Structural Cluster Statistic (SCS)**. SCS quantifies structural autocorrelation among prediction errors, effectively identifying structurally clustered error patterns. This metric complements existing evaluation methods by providing deeper insights into structurally predictive behaviors, thereby improving both model selection and interpretability in network-structured tasks.

• Beyond evaluation, we demonstrate how our SCS metric can be adapted into a regularization framework during model training. Specifically, we introduce the **Structural-Cluster-Aware (SCA)** learning objective, an extension of SCS designed to explicitly regularize structural error distributions. Incorporating SCA encourages GNNs to yield predictions with fewer structurally clustered errors, which is especially advantageous for critical network/graph applications requiring robust and reliable performance, such as real-time fault-tolerant network systems.

• We extensively validate our proposed metric and regularization approach through empirical studies involving multiple benchmark and synthetic datasets, as well as representative GNN architectures. Our results yield several important findings: (1) existing GNN models consistently exhibit structurally clustered prediction errors, highlighting the inadequacy of traditional evaluation metrics; (2) distinct structural error patterns are significantly influenced by the underlying graph structure (e.g., regular versus power-law connectivity) rather than by architectural differences among GNN variants; (3) our SCA regularization method effectively mitigates structural error clustering, significantly enhancing the structural robustness and reliability of GNN predictions.

## 2 PRELIMINARY AND BACKGROUND

**Graph Data and Network Applications.** A graph is formally defined as $\mathcal{G} = (\mathcal{V}, \mathcal{E})$, where $\mathcal{V} = \{v_1, v_2, \ldots, v_n\}$ denotes the set of nodes (vertices), and $\mathcal{E} \subseteq \mathcal{V} \times \mathcal{V}$ represents the set of edges that capture relationships or interactions between nodes. Each node $v \in \mathcal{V}$ is associated with a feature vector $\mathbf{x}_v \in \mathbb{R}^d$, encoding relevant attributes or measurements specific to that node. In network applications, prediction tasks are commonly defined at the node level, with each node assigned a label $y_v \in \mathcal{Y}$. For instance, in environmental monitoring scenarios, the goal might be to classify each node based on pollution intensity (e.g., low, moderate, or high) using node-specific measurements. The primary objective is therefore to learn a predictive model that effectively integrates local node features and graph topology to accurately predict individual node labels. Additionally, we denote the adjacency matrix as $\mathbf{A}$, the degree matrix as $\mathbf{D}$ and the graph Laplacian and normalized Laplacian matrices as $\mathbf{L} = \mathbf{D} - \mathbf{A}$ and $\widehat{\mathbf{L}} = \mathbf{D}^{-1/2}\mathbf{L}\mathbf{D}^{-1/2}$, respectively.

**Graph Neural Networks (GNNs).** GNN models broadly fall into three families based on their approaches for capturing graph structures: *message-passing (spatial-based), spectral-based, and graph transformer* methods (see Appendix A for a more detailed introduction and discussion). Message-passing GNNs capture dependencies by iteratively aggregating and updating node embeddings based on local neighborhood information. These models naturally encode local graph structures without explicit spectral decomposition, making them computationally efficient and intuitive. Spectral-based GNNs define graph convolution operations via spectral filtering based on the eigen-decomposition of the graph Laplacian. They effectively capture global graph structures but often require computationally efficient approximations due to complexity constraints. Graph transformers extend the spatial approach by incorporating self-attention mechanisms, dynamically adapting the weights of neighboring nodes. This method allows flexible modeling of both local and global dependencies and often uses positional encodings to further enhance node representations.

**Performance Evaluation of GNNs.** The performance evaluation of GNN models is typically carried out on a dedicated test set, denoted as $\mathcal{V}_{\text{test}} = \{v_1, v_2, \ldots, v_k\} \subset \mathcal{V}$. Ground-truth labels for these nodes are represented as $\mathcal{Y}_{\text{test}} = \{y_1, y_2, \ldots, y_k\}$, while the predictions from the GNN model are denoted by $\widehat{\mathcal{Y}} = \{\widehat{y}_1, \widehat{y}_2, \ldots, \widehat{y}_k\}$. The evaluation involves quantifying discrepancies between predicted and true labels using an appropriate performance metric. Formally, given a complete vector of ground-truth labels $\mathbf{Y} = [y_1, y_2, \ldots, y_k]$ and corresponding predictions $\widehat{\mathbf{Y}} = [\widehat{y}_1, \widehat{y}_2, \ldots, \widehat{y}_k]$, we define the error vector $\boldsymbol{\epsilon}$ as:

$$\boldsymbol{\epsilon} = (\epsilon_1, \ldots, \epsilon_k), \quad \epsilon_i = f(y_i, \widehat{y}_i),$$

where $f(\cdot)$ denotes a pointwise loss function that measures the deviation or severity of the prediction error. Typical examples include the square error $f(y_i, \widehat{y}_i) = (y_i - \widehat{y}_i)^2$ for regression tasks, or the misclassification indicator $f(y_i, \widehat{y}_i) = \mathbb{I}(y_i \neq \widehat{y}_i)$ for classification tasks.

## 3 MAIN RESULTS

In this section, we present our main results. We first introduce a mathematical framework that allows us to formally analyze the capability of evaluation metrics, highlighting critical limitations inherent in commonly used metrics. Based on these insights, we propose a novel structure-aware evaluation metric inspired by Moran's I statistic Moran (1950), and demonstrate how it can also serve as a regularizer during the training process.

### 3.1 LIMITATIONS OF CURRENT EVALUATION METRICS

We first introduce a formal framework to rigorously characterize the capability of evaluation metrics to differentiate distinct structural error patterns. Inspired by the concept of expressive power in GNN literature, we define the expressiveness of an evaluation metric as its ability to distinguish between different error distributions in expectation:

**Definition 1** (Distribution Expressiveness of Evaluation Metrics)**.** *Let $\mu(\cdot)$ be an evaluation metric mapping the error vector $\boldsymbol{\epsilon}$ to a real value. Given two distinct error distributions $\mathbb{P}_1$ and $\mathbb{P}_2$, we say the evaluation metric $\mu$ can differentiate between these distributions if $\mathbb{E}_{\mathbb{P}_1}[\mu(\boldsymbol{\epsilon})] \neq \mathbb{E}_{\mathbb{P}_2}[\mu(\boldsymbol{\epsilon})]$.*

This definition allows us to formally evaluate how well existing metrics capture meaningful differences in structural error distributions. Next, we introduce a key property commonly exhibited by traditional metrics used in evaluating GNN performance.

**Definition 2** (Exchangeable Measure). *For a graph with $N$ nodes, an evaluation metric $\mu(\cdot)$ is said to be* exchangeable *if, for any permutation $\pi$ of node indices $\{1, 2, \ldots, N\}$, it holds that:*

$$\mu(\boldsymbol{\epsilon}) = \mu(\pi(\boldsymbol{\epsilon})), \quad where \quad \pi(\boldsymbol{\epsilon}) = [\epsilon_{\pi(1)}, \epsilon_{\pi(2)}, \ldots, \epsilon_{\pi(N)}].$$

Intuitively, exchangeability implies that the metric's evaluation does not depend on the ordering of errors but only on their multiset. Conventional metrics for evaluating GNN performance (e.g., ACC, MSE, F1-score, AU-ROC) satisfy this definition. For instance, ACC, defined as:

$$\mu_{\text{ACC}}(\boldsymbol{\epsilon}) = \frac{1}{N} \sum_{i=1}^{N} \mathbb{I}(\epsilon_i = 0),$$

where $\mathbb{I}(\cdot)$ is an indicator function, clearly remains unchanged under any permutation $\pi$ of the node indices:

$$\mu_{\text{ACC}}(\boldsymbol{\epsilon}) = \frac{1}{N} \sum_{i=1}^{N} \mathbb{I}(\epsilon_i = 0) = \frac{1}{N} \sum_{i=1}^{N} \mathbb{I}(\epsilon_{\pi(i)} = 0) = \mu_{\text{AP}}(\pi(\boldsymbol{\epsilon})).$$

Thus, ACC is exchangeable, and similar reasoning can be applied to other common evaluation metrics. However, exchangeable metrics inherently face critical limitations, which we formalize in the following theorem:

**Theorem 3.1** (Limitation of Exchangeable Metrics). *Let $\mathbb{P}_1$ and $\mathbb{P}_2$ be two distinct error distributions for a GNN on a given graph $\mathcal{G}$. Suppose $\mu(\cdot)$ is an exchangeable evaluation metric. Then,*

$$\mathbb{E}_{\mathbb{P}_1}[\mu(\boldsymbol{\epsilon})] = \mathbb{E}_{\mathbb{P}_2}[\mu(\boldsymbol{\epsilon})], \quad provided\ that \quad \mathbb{E}_{\mathbb{P}_1}[S(\boldsymbol{\epsilon})] = \mathbb{E}_{\mathbb{P}_2}[S(\boldsymbol{\epsilon})],$$

*where $S(\boldsymbol{\epsilon}) = \sum_{v \in \mathcal{V}} f(y_v, \widehat{y}_v)$.*

Theorem 3.1 indicates that exchangeable metrics cannot distinguish between error patterns if the total magnitude or frequency of errors is identical, regardless of how those errors are distributed over the graphs. These metrics treat errors merely as interchangeable entities, failing to account for their topological arrangement on the graph. Consequently, such metrics are insufficient for evaluating GNNs predictive performance, especially in applications where the structure of errors carries critical information.

**Concrete Examples.** To illustrate these theoretical limitations more concretely, consider two models predicting traffic flow in a network, with different error distributions. In the first model, errors are uniformly distributed across the network with larger individual errors. In the second model, smaller errors occur, but they are concentrated at specific critical locations. Traditional metrics, such as MSE, would favor the second model due to its lower average error. However, these metrics fail to capture the impact of error distribution. Despite smaller average errors, the clustering of errors in critical areas (e.g., congestion points) can have severe consequences. In contrast, the first model, though it has larger individual errors, distributes them evenly, leading to a lower risk of localized failures. This example demonstrates how traditional metrics can overlook critical issues by not considering the structural distribution of errors. For a detailed quantitative example, see Appendix D.1.

Figure 1 provides a visual illustration, where scenarios exhibit identical counts of correct and incorrect predictions (i.e., same ACC) but vary significantly in structural error patterns, ranging from clustered to dispersed distributions. As discussed, this variability is critically important in practical applications such as traffic management or environmental monitoring, where clustered errors demand urgent attention. The inability of traditional exchangeable metrics to differentiate these structurally distinct patterns highlights substantial risks associated with model evaluation and deployment decisions based solely on conventional metrics.

## 3.2 STRUCTURAL CLUSTERING STATISTICS (SCS)

As illustrated, traditional evaluation metrics typically treat prediction errors independently, neglecting the structural relationships and thus failing to differentiate between randomly distributed errors and meaningful structural clusters. To address this, we need a metric, which explicitly accounts for node-to-node relationships within the graph.

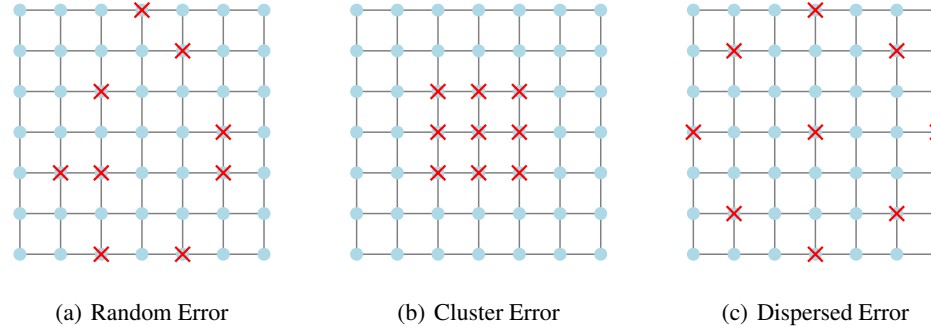

(a) Random Error  (b) Cluster Error  (c) Dispersed Error

Figure 1: An illustration of distinct structural error patterns. Figure 1(a) (*Random Error*): Incorrect predictions (marked by red crosses) are randomly dispersed across the graph. Figure 1(b) (*Cluster Error*): Incorrect predictions are concentrated within a localized region of the graph. Figure 1(c) (*Dispersed Error*): Incorrect predictions are evenly spaced and distributed apart from one another across the graph. These differing patterns underscore the necessity of using structure-aware evaluation metrics when assessing GNN predictions.

**Limitations of Naive Structural Metrics.** A straightforward structural evaluation method might measure the structural pattern of errors using average shortest-path distances between incorrectly predicted nodes. However, this naive approach has significant practical limitations. First, it is computationally expensive, particularly for large graphs. Moreover, in graphs with irregular or hub-based structures (e.g., power-law distributed graphs), highly connected nodes disproportionately influence distance-based metrics, obscuring genuine structural clustering patterns and limiting interpretability Barabási (2013).

**Formulation of SCS.** To address these shortcomings, we introduce the SCS metric, inspired by Moran's I statistic. SCS quantifies structural autocorrelation by measuring how prediction errors at each node correlate with those of neighboring nodes. This property makes it particularly effective for identifying structural clusters of prediction errors.

Formally, let the prediction error at a test node $v \in \mathcal{V}_{\text{test}}$ be defined as $\epsilon_v = f(y_v, \widehat{y}_v)$, where $y_v$ is the ground-truth label and $\widehat{y}_v$ is the predicted label generated by the GNN. SCS, computed exclusively over the test set $\mathcal{V}_{\text{test}}$, is given by:

$$\text{SCS}(\boldsymbol{\epsilon}, \mathcal{V}_{\text{test}}) = \frac{k}{W} \frac{\sum_{v,u \in \mathcal{V}_{\text{test}}} w_{vu}(\epsilon_v - \bar{\epsilon})(\epsilon_u - \bar{\epsilon})}{\sum_{v \in \mathcal{V}_{\text{test}}} (\epsilon_v - \bar{\epsilon})^2}, \quad (3.1)$$

where $\bar{\epsilon} = 1/k \sum_{v \in \mathcal{V}_{\text{test}}} \epsilon_v$ denotes the mean prediction error across all test nodes, $w_{vu}$ represents the connection weight, $W = \sum_{v,u \in \mathcal{V}_{\text{test}}} w_{vu}$ is the sum of all connection weights over the test set, and $k = |\mathcal{V}_{\text{test}}|$ is the number of nodes in the test set.

SCS explicitly quantifies the correlation of prediction errors among neighboring nodes within the test set. A positive SCS indicates structurally clustered errors, revealing localized regions where the model fails to accurately capture structural dependencies. Values close to zero indicate randomly distributed errors, while negative values imply structurally dispersed error patterns, highlighting discrepancies between model predictions and underlying graph structures. By employing this adapted SCS, we obtain a precise, interpretable, and computationally efficient measure of predictive performance over the graph, thus facilitating targeted model improvements by pinpointing test regions where GNN models exhibit poor structural predictive capabilities.

### 3.3 STRUCTURAL-CLUSTER-AWARE (SCA) REGULARIZATION

In previous sections, we introduced the SCS as a robust metric for effectively detecting and quantifying structurally clustered errors in GNN predictions. While identifying these error clusters is essential, from a practical standpoint, promoting structurally uniform error distributions is equally critical for real-world network applications. Specifically, structurally correlated prediction errors can propagate rapidly within real-time and fault-tolerant systems—such as environmental sensor

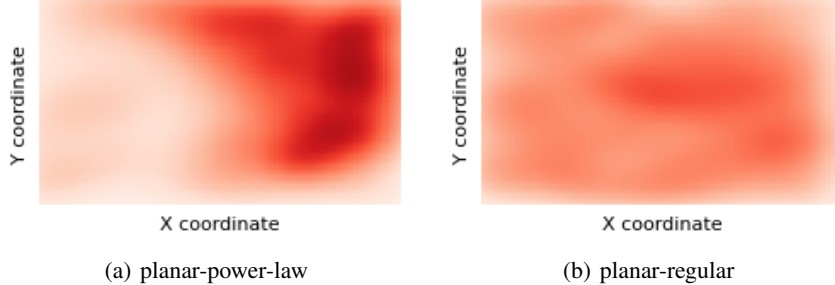

(a) planar-power-law            (b) planar-regular

Figure 2: An illustration of structural error patterns in graphs with different structures. The $x$ and $y$ axes represent the coordinates of nodes, and the intensity of the red colour indicates the severity of prediction errors. As shown, the planar-power-law (left) exhibits pronounced structurally clustered errors, whereas the planar-regular (right) displays a more dispersed, uniformly distributed error.

networks, smart grids, and critical infrastructure monitoring—potentially leading to biased predictions, localized failures, and suboptimal decision-making (Zhao et al., 2019b; Chu & Sethu, 2010). Therefore, an important question arises: *can we leverage our proposed structural metric to encourage GNN models to produce more uniformly distributed prediction errors?*

Uniform error distributions offer significant practical benefits. For instance, in environmental monitoring, uniformity reduces the likelihood of systematic regional biases, thereby enhancing reliability, fairness, and accuracy in environmental assessments and policy-making. Similarly, for sensor network deployments, structural uniformity in prediction errors mitigates localized blind spots or overly concentrated error regions, leading to improved overall system resilience and balanced performance across the entire graph domain (Chu & Sethu, 2010).

However, directly employing the original SCS formulation as a regularizer presents notable optimization challenges. Specifically, the original metric can yield negative and unbounded values, complicating gradient-based training and potentially causing numerical instability, especially when prediction variances are small. Minimizing negative structural autocorrelation might inadvertently promote dispersed error patterns rather than uniformity, contradicting the desired optimization goal.

To overcome these limitations, we propose a modified regularization term, Structural-Cluster-Aware (SCA) regularizer. This adjusted form, based on a squared version of the SCS metric, ensures the regularization term is always non-negative and bounded, effectively penalizing significant structural autocorrelation (either clustered or dispersed). Formally, the SCA regularization term is defined as:

$$\mathcal{L}_{\text{SCA}}(\boldsymbol{\epsilon}, \mathcal{G}, \delta) = \left( \frac{k}{W} \frac{\sum_{i,j} w_{ij}(\epsilon_i - \bar{\epsilon})(\epsilon_j - \bar{\epsilon})}{\sum_i (\epsilon_i - \bar{\epsilon})^2 + \delta} \right)^2, \tag{3.2}$$

where all variables are as previously defined. We introduce a small positive constant $\delta$ in the denominator to ensure numerical stability, particularly in situations where prediction variance is low, thereby avoiding potential division by zero. In practice, $\delta$ is typically chosen as a very small value (e.g., $10^{-6}$), minimally influencing the regularization objective while effectively preventing numerical instabilities. The squared formulation of the SCA regularizer guarantees non-negativity, explicitly penalizing significant structural autocorrelation (whether clustered or dispersed). This approach effectively encourages structural uniformity in prediction errors and mitigates localized error clustering. Integrating this structure-aware regularization into the total loss function results in:

$$\mathcal{L}_{\text{total}} = \mathcal{L}_{\text{task}}(\mathbf{Y}, \widehat{\mathbf{Y}}) + \lambda \mathcal{L}_{\text{SCA}}(\boldsymbol{\epsilon}, \mathcal{G}, \delta), \tag{3.3}$$

where $\lambda \geq 0$ is a hyperparameter controlling the strength of the structural regularisation. Employing this regularizer encourages structurally consistent predictions, enhances robustness to faults (cluster errors), and ultimately improves the reliability of GNN systems deployed in real-time environments.

## 4 EMPIRICAL STUDY

In this section, we present an empirical study to investigate the structural characteristics of prediction errors in GNN models. Specifically, we aim to address the following key research questions:

| Dataset | Cora | | Citeseer | | California Housing | | Air Temperature | |
|---|---|---|---|---|---|---|---|---|
| Model | SCS ↓ | ACC(%) ↑ | SCS ↓ | ACC(%) ↑ | SCS ↓ | MSE ↓ | SCS ↓ | MSE ↓ |
| GCN | 0.21±0.03 | 84.4±0.5 | 0.23±0.03 | 79.2±0.6 | 0.14±0.02 | 0.038±3e-3 | 0.15±0.03 | 0.031±2e-3 |
| GCN-SCA | **0.10±0.01** | 84.0±0.4 | **0.11±0.02** | 78.8±0.5 | **0.10±0.01** | 0.040±2e-3 | **0.09±0.01** | 0.033±2e-3 |
| GraphSAGE | 0.19±0.02 | 88.5±0.4 | 0.20±0.03 | 80.1±0.4 | 0.13±0.03 | 0.041±2e-3 | 0.16±0.02 | 0.029±2e-3 |
| GraphSAGE-SCA | **0.11±0.01** | 88.2±0.3 | **0.13±0.02** | 79.7±0.4 | **0.08±0.01** | 0.042±2e-3 | **0.10±0.01** | 0.030±1e-3 |
| GAT | 0.18±0.03 | 88.8±0.3 | 0.19±0.02 | 80.4±0.5 | 0.11±2e-3 | 0.036±2e-3 | 0.14±0.01 | 0.027±1e-3 |
| GAT-SCA | **0.08±0.01** | 88.6±0.4 | **0.09±0.01** | 80.1±0.4 | **0.07±0.01** | 0.040±1e-3 | **0.09±0.01** | 0.029±1e-3 |
| ChebNet | 0.25±0.02 | 86.5±0.4 | 0.26±0.02 | 77.8±0.5 | 0.20±0.02 | 0.043±2e-3 | 0.19±0.03 | 0.032±2e-3 |
| ChebNet-SCA | **0.15±0.01** | 86.2±0.3 | **0.18±0.01** | 77.4±0.4 | **0.12±0.01** | 0.044±2e-3 | **0.14±0.02** | 0.033±2e-3 |
| GraphTransformer | 0.15±0.02 | 89.0±0.3 | 0.16±0.02 | 81.2±0.4 | 0.18±0.02 | 0.035±2e-3 | 0.18±0.02 | 0.027±1e-3 |
| GraphTransformer-SCA | **0.07±0.01** | 88.7±0.3 | **0.10±0.01** | 80.9±0.4 | **0.11±0.01** | 0.035±1e-3 | **0.12±0.01** | 0.028±1e-3 |
| Avg. Improvement (%) | 47.96 | | 41.35 | | 36.84 | | 32.92 | |

Table 1: Evaluation of GNN models with and without the proposed SCA regularization across classification (Cora, Citeseer) and regression (California Housing, Air Temperature) datasets. Models labeled '-SCA' indicate training with our proposed regularization. Lower values (↓) indicate better performance for SCS and Mean Squared Error (MSE), while higher values (↑) indicate better performance for accuracy (ACC). Bold entries highlight improvements achieved by incorporating SCA. The final row summarizes the average percentage of SCS improvement across each dataset, illustrating that SCA effectively mitigates structural clustering in prediction errors.

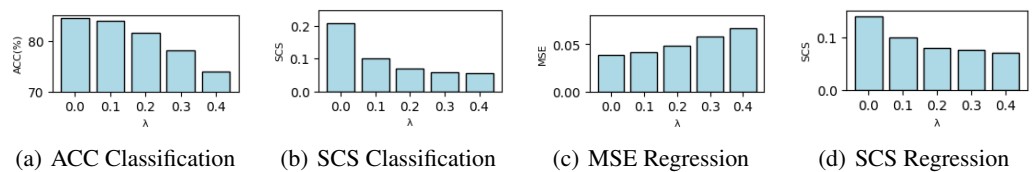

(a) ACC Classification  (b) SCS Classification  (c) MSE Regression  (d) SCS Regression

Figure 3: An illustration of the effects of the regularization hyperparameter $\lambda$ on classification and regression tasks. Figures 3(a) and 3(b) show that increasing $\lambda$ reduces structural clustering of errors (SCS) but simultaneously decreases classification accuracy (ACC). Similarly, Figures 3(c) and 3(d) illustrate that increasing $\lambda$ reduces structural clustering in regression errors (SCS) at the expense of increased MSE. Thus, $\lambda$ effectively manages the trade-off between structural uniformity of errors and overall predictive performance.

1. Do existing GNN models exhibit structurally clustered prediction errors, and if so, how do these clusters differ among various GNN architectures?

2. Is the proposed SCS effective in identifying structurally clustered errors?

3. Can SCA learning effectively mitigate structural clusters in errors?

Implementation details and hyperparameter selections are deferred to the supplementary material.

## 4.1 Experimental Design

**Datasets and Baselines.** To comprehensively evaluate our proposed methods across diverse contexts, we select benchmark datasets covering both node classification and regression tasks. Specifically, we use two widely adopted citation network datasets for node classification tasks: *Cora* and *Citeseer*(Sen et al., 2008). For node regression tasks, we use two spatially structured real-world datasets: the *California Housing Prices* datasetPace & Barry (1997) and the *Air Temperature* dataset Hooker et al. (2018). Additionally, to investigate how underlying graph structures influence model performance, we synthesize two planar graph regression datasets characterized by distinct structural patterns: *planar-regular* (uniform degree distribution) and *planar-power-law* (power-law degree distribution). The procedure for synthesis is provided in the supplementary material. For GNNs, we select five representative models covering three prominent categories: message-passing-based GNNs (*GCN*(Kipf & Welling, 2017), *GraphSAGE*(Hamilton et al., 2017), and *GAT*(Veličković et al., 2017)), spectral-based GNNs (*ChebNet*(Defferrard et al., 2016)), and graph transformers (*Graph Transformer*(Dwivedi & Bresson, 2020)). The implementations for these

baselines follow widely adopted repositories(Dwivedi et al., 2023; Dwivedi & Bresson, 2020; Hu et al., 2020), employing standard training procedures and hyperparameter tuning strategies based on validation sets to ensure fairness.

**Evaluation Tasks and Metrics.** We conduct evaluation across two distinct prediction tasks: node classification and node regression. For classification datasets, we measure performance using ACC, while for regression datasets, we employ MSE. To evaluate the structural characteristics of prediction errors, we use the proposed SCS. Each GNN model is evaluated under two conditions: with and without our proposed SCA regularization. For all datasets, we adopt either default data splits (where provided) or apply a standard random split with a 60%/20%/20% ratio for training, validation, and testing subsets, respectively. All reported results are averaged over five independent trials, ensuring statistical robustness and reproducibility.

### 4.2 Experimental Results

**Structural Error Patterns.** Our first set of experiments investigates whether existing GNNs produce structurally clustered prediction errors and evaluates if our proposed SCS effectively captures these patterns. Figure 2 illustrates representative structural error distributions on graphs with different underlying structures. Notably, we observe that the graph structure itself, rather than specific GNN architectures, primarily determines the structural distribution of prediction errors. Specifically, planar graphs with power-law connectivity tend to exhibit pronounced structurally clustered errors, while planar graphs with regular connectivity display more uniformly dispersed errors. Given that many real-world graphs, such as those found in sensor networks or urban infrastructures, often exhibit scale-free (power-law) characteristics Barabási (2013), our findings suggest that GNNs deployed in these contexts will typically produce structurally clustered prediction errors. This observation is quantitatively confirmed by positive SCS values across all evaluated GNN models (Table 1). The consistency of these results across different architectures further emphasizes that structural clustering is an intrinsic property related to the underlying graph topology rather than being driven solely by model-specific factors. These findings not only highlight the critical importance of explicitly considering error structures on the graph in evaluation but also validate the effectiveness of our proposed SCS metric in identifying and quantifying structural clustering.

**Effectiveness of SCA.** We next assess the effectiveness of our proposed SCA regularization method. Table 1 clearly demonstrates that incorporating SCA significantly reduces structural clustering of prediction errors (indicated by consistently lower SCS values) across all GNN architectures and datasets. Particularly notable improvements occur in message-passing-based models such as GCN, GraphSAGE, and GAT, where structural clustering is substantially reduced by approximately 40%-48%, with only minor degradations in predictive performance (ACC or MSE). Transformer-based and spectral-based models exhibit slightly smaller reductions, likely due to their inherently less-clustered baseline error distributions. Nevertheless, these improvements underscore the broad practical effectiveness of the SCA regularization, particularly valuable for structurally sensitive real-world applications like environmental monitoring and sensor networks.

**Hyperparameter Analysis.** Finally, we conduct a sensitivity analysis on the regularization hyperparameter $\lambda$, which controls the intensity of the SCA objective. As illustrated in Figure 3, increasing $\lambda$ systematically reduces structural clustering of errors (lower SCS) for both classification (Figure 3(b)) and regression (Figure 3(d)) tasks. However, these improvements in structural uniformity come at a slight cost to predictive accuracy, as demonstrated by decreased ACC scores for classification (Figure 3(a)) and increased MSE for regression (Figure 3(c)). Consequently, $\lambda$ acts as a trade-off parameter balancing the structural uniformity of prediction errors against overall predictive accuracy. Empirically, we identify $\lambda = 0.1$ as a favorable setting for our settings, achieving a substantial reduction in structural clustering without significantly compromising predictive performance, as summarized in Table 1.

## 5 Related Works

Graph representation learning has received substantial attention recently, driven by the increasing necessity to effectively analyze complex relational structures embedded within graph data (see com-

prehensive surveys by Hamilton (2020); Kazemi et al. (2020)). Among various approaches, Graph Neural Networks (GNNs) have proven particularly effective, achieving state-of-the-art performance across diverse graph-related tasks, notably in spatial applications such as traffic forecasting, urban planning, and environmental monitoring (Jiang & Luo, 2022; Dong et al., 2023; Zhang et al., 2023). Broadly, existing GNN architectures can be categorized into three main classes based on their structural learning approaches: (1) *message-passing methods*, which aggregate local neighborhood information to capture immediate graph connectivity (Kipf & Welling, 2017; Xu et al., 2020; Veličković et al., 2017); (2) *spectral-based methods*, leveraging graph Laplacian eigen-decompositions to encode global structural information (Defferrard et al., 2016; Bruna et al., 2013); and (3) *graph transformer models*, utilizing self-attention mechanisms to model long-range node interactions and dependencies (Dwivedi & Bresson, 2020). Given their practical significance, extensive research efforts have focused on theoretical foundations (Jegelka, 2022; Bronstein et al., 2021), architectural innovations (Wu et al., 2020), and computational optimizations for efficient training and inference of GNNs (Shao et al., 2024; Fey et al., 2021).

Despite these advancements, recent studies emphasize that inadequate evaluation methodologies remain a crucial barrier hindering further progress in the GNN field (Bechler-Speicher et al., 2025; Shchur et al., 2018). Rigorous and reproducible evaluation frameworks have become recognized as essential components of trustworthy and robust machine learning research, directly impacting model selection and practical deployment (Zhang et al., 2021; Pineau et al., 2021). Benchmarking studies in graph representation learning have comprehensively evaluated GNN performance across diverse hyperparameter configurations and learning paradigms, highlighting that evaluation outcomes can differ significantly under inductive versus transductive settings, as well as various temporal scenarios (Dwivedi et al., 2023; Dong et al., 2024; Errica et al., 2019; Hu et al., 2020; Lv et al., 2021). Furthermore, a growing body of literature has extended these evaluations to temporal domains, assessing the effectiveness of temporal GNN models in dynamic graph settings (Junuthula et al., 2018; Haghani & Keyvanpour, 2019; Poursafaei et al., 2022; Huang et al., 2024; Su & Wu, 2025).

Nevertheless, a crucial yet underexplored area in current research is the evaluation metrics of GNNs, particularly regarding structural error distributions. This paper addresses this gap by proposing a novel evaluation metric explicitly designed to capture and quantify structural clustering patterns in GNN prediction errors, providing deeper insights for model assessment and deployment.

## 6 CONCLUDING DISCUSSION

**Conclusion.** In this paper, we investigated evaluation metrics tailored for GNNs. We identified key limitations of conventional exchangeable metrics—such as ACC and MSE—in capturing essential structural error patterns, particularly the distinction between clustered and randomly dispersed prediction errors. To address these limitations, we proposed SCS, a novel structure-aware evaluation metric, explicitly designed to quantify structural clustering in prediction errors. Additionally, we introduced an extension of this metric, termed SCA learning, which serves as a regularizer to mitigate structurally clustered errors during GNN training. Our extensive empirical evaluation confirmed that the proposed metric provides deeper insights into structural error distributions, effectively distinguishing among different structural error patterns and improving both model selection and robustness in network structured tasks.

**Limitations and Future Work.** An intriguing observation from our empirical study is the strong relationship between structural error patterns and underlying structural properties of graphs, such as degree distributions. This correlation likely emerges because GNN predictions inherently depend on the provided graph topology. Further research is needed to thoroughly investigate this phenomenon and elucidate the precise mechanisms through which graph structure influences GNN predictive behaviors. Additionally, while our current framework focuses on structural error clustering, incorporating temporal dynamics or other structural patterns into this evaluation approach represents a promising direction for future work. We provide an extended discussion of these potential extensions in Appendix D.2. Integrating temporal dimensions would significantly enhance the applicability and robustness of our framework, enabling more comprehensive evaluations for GNNs in spatial-temporal contexts.

ETHICS STATEMENT

We have thoroughly reviewed the relevant ethical guidelines and confirm that this work does not raise any significant ethical concerns. Our study does not involve human participants, sensitive or proprietary data, or applications with foreseeable potential for harm. The methods and contributions presented adhere to the principles of fairness, transparency, and research integrity.

REPRODUCIBILITY STATEMENT

This paper provides complete disclosure of all necessary details required to reproduce the main experimental results. Specifically, we include explicit descriptions of the training and testing procedures, along with all relevant assumptions. The datasets used in this work are either publicly available or synthetically generated with detailed explanations, ensuring reproducibility and clarity of the empirical findings.

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

## THE USE OF LARGE LANGUAGE MODELS (LLMS)

We used LLMs only as a general-purpose writing assistant to aid in grammar checking and polishing the writing. The LLM did not contribute to research ideas, experiment design, theoretical analysis, or result interpretation.

## A DIFFERENT GRAPH NEURAL NETWORKS FAMILIES

Graph Neural Networks (GNNs) have become one of the most effective machine learning methods for modeling relational and spatial data due to their powerful ability to encode complex structural dependencies. Based on the approach used to capture graph dependencies, GNN architectures can broadly be categorized into three main families: *spatial-based (message-passing) GNNs*, *spectral-based GNNs*, and *graph transformers*. In this section, we provide a comprehensive introduction to each of these GNN families, highlighting their theoretical foundations, advantages, and practical considerations.

### A.1 SPATIAL-BASED (MESSAGE-PASSING) GNNS

Spatial-based GNNs, also known as message-passing GNNs, operate directly on the graph structure by iteratively aggregating and updating node representations based on their local neighborhoods. Unlike spectral methods, they do not require eigen-decomposition of graph matrices, making them computationally efficient and highly scalable for large graphs.

Formally, the general message-passing paradigm for updating the embedding of node $v$ at layer $t$ can be expressed as:

$$\mathbf{h}_v^{(t)} = \text{UPDATE}^{(t)}\left(\mathbf{h}_v^{(t-1)}, \text{AGGREGATE}^{(t)}\left(\{\mathbf{h}_u^{(t-1)} : u \in \mathcal{N}(v)\}\right)\right), \quad\quad \text{(A.1)}$$

where $\mathbf{h}_v^{(t)}$ is the embedding of node $v$ at layer $t$, and $\mathcal{N}(v)$ denotes its immediate neighbors. Representative models in this category include Graph Convolutional Networks (GCN) (Kipf & Welling, 2017), GraphSAGE (Hamilton et al., 2017), and Graph Attention Networks (GAT) (Veličković et al., 2017). Spatial-based GNNs naturally encode local structural information and gradually capture broader structural context as the network depth increases. However, excessively deep message-passing architectures often suffer from oversmoothing, where node representations converge, reducing their discriminative power.

### A.2 SPECTRAL-BASED GNNS

Spectral-based GNNs leverage graph spectral theory and define graph convolutions using spectral filtering based on eigen-decomposition of the graph Laplacian. Specifically, given the normalized Laplacian matrix $\widehat{\mathbf{L}} = \mathbf{U}\mathbf{\Lambda}\mathbf{U}^\top$, where $\mathbf{U}$ represents eigenvectors and $\mathbf{\Lambda}$ is a diagonal matrix of eigenvalues, the spectral convolution operation on node features $\mathbf{x}$ with a parameterized filter $g_\theta(\cdot)$ is defined as:

$$\mathbf{x} * \mathbf{g}_\theta = \mathbf{U} g_\theta(\mathbf{\Lambda}) \mathbf{U}^\top \mathbf{x}. \quad\quad \text{(A.2)}$$

Early spectral-based GNNs explicitly computed the eigen-decomposition, leading to significant computational complexity. To address this limitation, practical implementations such as Chebyshev networks (ChebNet) (Defferrard et al., 2016) and the simplified Graph Convolutional Networks by Kipf & Welling (2017) use polynomial approximations, significantly enhancing computational efficiency. While spectral-based methods effectively capture global structural properties of graphs, their reliance on spectral decomposition makes them inherently less scalable for large-scale graph datasets compared to spatial-based approaches.

### A.3 GRAPH TRANSFORMERS

Graph transformers extend the spatial-based message-passing framework by incorporating self-attention mechanisms, allowing models to adaptively weigh information from nodes across varying distances within the graph. Inspired by transformer architectures initially developed for natural

language processing, graph transformers apply attention mechanisms directly to graph structures to capture both local and long-range dependencies.

Formally, given node embeddings $\mathbf{h}_v$ and $\mathbf{h}_u$, the attention mechanism computes attention coefficients $\alpha_{vu}$ between nodes $v$ and $u$ as follows:

$$\alpha_{vu} = \frac{\exp\left(\text{Attn}(\mathbf{h}_v, \mathbf{h}_u)\right)}{\sum_{k \in \mathcal{V}} \exp\left(\text{Attn}(\mathbf{h}_v, \mathbf{h}_k)\right)}, \tag{A.3}$$

where $\mathcal{V}$ represents the set of nodes. Unlike traditional message-passing approaches, graph transformers can dynamically and selectively attend to neighbors at varying graph distances, making them highly effective in modeling complex spatial interactions. To incorporate structural information explicitly, graph transformers typically use positional encodings derived from the graph structure, thereby augmenting node feature representations. Despite their ability to capture richer representations and dependencies, graph transformers typically require more computational resources, especially for larger graphs, due to the quadratic complexity associated with computing pairwise attention scores.

## A.4 COMPARISON AND MOTIVATION

Empirical comparisons among spatial-based GNNs, spectral-based GNNs, and graph transformers often indicate similar overall predictive performances (measured by metrics like accuracy) across various datasets. The primary differences between these families generally manifest in their trade-offs regarding computational efficiency and the scope of structural information captured. Specifically, spatial-based methods offer scalability and efficient local aggregation but may have difficulty encoding global structures effectively without increased depth. Spectral-based methods explicitly encode global structure but can be computationally prohibitive for large-scale graphs. Graph transformers flexibly capture both local and global dependencies but at a higher computational cost.

The subtle performance differences and limited understanding of each family's capability to capture specific graph structures underscore the importance of systematically evaluating and understanding GNN models. This necessity motivates our work in this paper—focusing on the development of spatially-aware evaluation metrics capable of revealing nuanced differences in GNN performance, particularly in spatial applications.

## B PROOFS

In this appendix, we present a proof for Theorem 3.1.

**Proof of Theorem 3.1.** Recall from Definition 2 that an evaluation metric $\mu(\cdot)$ is exchangeable if, for any permutation $\pi$ of node indices, the metric satisfies:

$$\mu(\boldsymbol{\epsilon}) = \mu(\pi(\boldsymbol{\epsilon})).$$

Let $\mathbb{P}_1$ and $\mathbb{P}_2$ be two distinct error distributions for a given GNN applied to a graph $\mathcal{G} = (\mathcal{V}, \mathcal{E})$, where $|\mathcal{V}| = N$. Suppose additionally:

$$\mathbb{E}_{\mathbb{P}_1}\left[\text{S}(\boldsymbol{\epsilon})\right] = \mathbb{E}_{\mathbb{P}_2}\left[\text{S}(\boldsymbol{\epsilon})\right],$$

where the sum-based metric is defined as:

$$\text{S}(\boldsymbol{\epsilon}) = \sum_{v \in \mathcal{V}} f(y_v, \widehat{y}_v).$$

Since $\mu(\cdot)$ is exchangeable, its evaluation depends solely on the multiset of error values $\{\epsilon_1, \epsilon_2, \ldots, \epsilon_N\}$ rather than on their spatial arrangement or indexing.

Notice that the metric $\text{S}(\boldsymbol{\epsilon})$ itself is inherently exchangeable, as it is simply a sum over nodes, invariant under permutations. Thus, the condition:

$$\mathbb{E}_{\mathbb{P}_1}\left[\text{S}(\boldsymbol{\epsilon})\right] = \mathbb{E}_{\mathbb{P}_2}\left[\text{S}(\boldsymbol{\epsilon})\right]$$

implies that the two distributions $\mathbb{P}_1$ and $\mathbb{P}_2$ yield identical expectations for every exchangeable aggregation of errors, as these aggregations remain invariant under any permutation.

Let $\mathcal{E}_m$ denote the set of all possible multisets of error values. Since $\mu(\cdot)$ is exchangeable, we can express its expectation under a given distribution $\mathbb{P}$ as:

$$\mathbb{E}_{\mathbb{P}}[\mu(\boldsymbol{\epsilon})] = \sum_{E \in \mathcal{E}_m} \mu(E)\mathbb{P}(E),$$

where $\mathbb{P}(E)$ represents the probability of observing the error multiset $E$.

Given the earlier equality for the sum-based aggregation, we have:

$$\mathbb{E}_{\mathbb{P}_1}[\mathrm{S}(\boldsymbol{\epsilon})] = \sum_{E \in \mathcal{E}_m} \mathrm{S}(E)\mathbb{P}_1(E) = \sum_{E \in \mathcal{E}_m} \mathrm{S}(E)\mathbb{P}_2(E) = \mathbb{E}_{\mathbb{P}_2}[\mathrm{S}(\boldsymbol{\epsilon})].$$

Because this equality holds for every exchangeable sum-based aggregation $\mathrm{S}(E)$, it follows directly that for each multiset $E$, we must have:

$$\mathbb{P}_1(E) = \mathbb{P}_2(E), \quad \forall E \in \mathcal{E}_m.$$

Therefore, since the evaluation metric $\mu(\cdot)$ is solely dependent on these multisets (due to exchangeability), we obtain:

$$\mathbb{E}_{\mathbb{P}_1}[\mu(\boldsymbol{\epsilon})] = \sum_{E \in \mathcal{E}_m} \mu(E)\mathbb{P}_1(E) = \sum_{E \in \mathcal{E}_m} \mu(E)\mathbb{P}_2(E) = \mathbb{E}_{\mathbb{P}_2}[\mu(\boldsymbol{\epsilon})].$$

Hence, we have formally shown that if two error distributions yield identical expectations for exchangeable sum-based aggregations, any exchangeable evaluation metric will fail to differentiate between these distributions. Thus, we establish the theorem statement:

$$\mathbb{E}_{\mathbb{P}_1}[\mu(\boldsymbol{\epsilon})] = \mathbb{E}_{\mathbb{P}_2}[\mu(\boldsymbol{\epsilon})],$$

as required. $\square$

### B.1 Limitations of Shortest-Path Distance and Advantages of SCS

Accurately evaluating spatial prediction errors in graph neural networks (GNNs) demands metrics that explicitly consider spatial relationships among nodes. Although an intuitive spatial measure might employ the average shortest-path distance between incorrectly predicted nodes, this naive metric faces significant practical and interpretative limitations. Consequently, we propose the Spatial Cluster Statistic (SCS), a robust metric that effectively captures spatial clustering by measuring spatial autocorrelation among prediction errors.

**Failure of Shortest-Path Distance Metrics.** A straightforward spatial evaluation approach involves computing the average shortest-path distance between nodes where prediction errors occur. At first glance, this method seems effective: smaller average distances might indicate spatially clustered errors, whereas larger average distances could reflect more dispersed errors. However, this approach exhibits several fundamental flaws:

- **Computational Complexity:** Shortest-path computations generally incur high computational costs, scaling poorly with network size. Typical algorithms such as Floyd-Warshall or multiple runs of Dijkstra's algorithm have time complexities of $O(N^3)$ and $O(N^2 \log N)$, respectively, making them impractical for large spatial networks or frequent evaluations **?**.

- **Distortion by Graph Structure:** In many real-world networks characterized by irregular connectivity or hub-like structures (such as scale-free graphs), shortest-path metrics are disproportionately influenced by high-degree nodes (hubs). Errors connected through hubs may exhibit artificially small distances despite being geographically distant, obscuring genuine spatial clustering patterns and limiting the interpretability of results Barabási (2013).

- **Ambiguity in Spatial Interpretation:** Shortest-path distances in graphs do not directly correspond to true spatial or geographical distances. Consequently, interpreting spatial patterns based solely on shortest-path measures can be misleading. Nodes that are physically far apart might have short graph distances due to high connectivity, while physically adjacent nodes could have long shortest-path distances if connections are sparse or irregular.

- **Inability to Identify Genuine Clustering:** Shortest-path metrics fail to distinguish between spatially meaningful clusters of errors and coincidental proximity caused by graph topology. Such metrics focus exclusively on distance magnitude, overlooking the crucial spatial autocorrelation (the correlation of errors between neighboring nodes), essential for identifying systematic spatial patterns.

Due to these critical limitations, shortest-path-based metrics are fundamentally inadequate for rigorously capturing spatial clustering in GNN prediction errors.

**Advantages and Interpretation of SCS.** To address these limitations, we introduce the Spatial Cluster Statistic (SCS), inspired by Moran's I statistic from spatial statistics. SCS explicitly accounts for spatial autocorrelation, quantifying how prediction errors at each node correlate with errors at neighboring nodes. Formally, given the prediction errors $\epsilon_v = f(y_v, \widehat{y}_v)$ for each node $v$ in a test set $\mathcal{V}_{\text{test}}$, SCS is defined as:

$$\text{SCS}(\boldsymbol{\epsilon}, \mathcal{V}_{\text{test}}) = \frac{k}{W} \frac{\sum_{v,u \in \mathcal{V}_{\text{test}}} w_{vu}(\epsilon_v - \bar{\epsilon})(\epsilon_u - \bar{\epsilon})}{\sum_{v \in \mathcal{V}_{\text{test}}}(\epsilon_v - \bar{\epsilon})^2}, \tag{B.1}$$

where:

- $\bar{\epsilon} = \frac{1}{k} \sum_{v \in \mathcal{V}_{\text{test}}} \epsilon_v$ is the mean error across test nodes;
- $w_{vu}$ are spatial weights, typically adjacency-based ($w_{vu} = 1$ if nodes $v$ and $u$ are neighbors, otherwise 0);
- $W = \sum_{v,u \in \mathcal{V}_{\text{test}}} w_{vu}$ represents the total weight sum;
- $k = |\mathcal{V}_{\text{test}}|$ is the number of test nodes.

SCS possesses several important advantages and clear interpretative properties:

- **Explicit Spatial Autocorrelation Measurement:** Unlike shortest-path metrics, SCS directly quantifies the spatial correlation of errors among neighboring nodes. Positive SCS values indicate pronounced spatial clustering of errors, revealing localized model inaccuracies. Conversely, negative values highlight dispersed error patterns, indicating that errors occur in a spatially repulsive manner.

- **Robustness to Graph Topology:** Because SCS evaluates autocorrelation based explicitly on neighborhood structures rather than shortest paths, it is inherently more robust to irregular graph structures and less distorted by highly connected nodes or hubs.

- **Computational Efficiency:** SCS computation is highly efficient and scalable ($O(E)$ complexity, where $E$ is the number of edges in the test subgraph), making it practical for repeated evaluation, hyperparameter tuning, and real-time monitoring of model performance on large-scale spatial networks.

- **Interpretability and Practical Insights:** SCS provides meaningful, actionable insights into spatial error structures. High positive values clearly indicate specific regions or node clusters needing model improvement or immediate attention, significantly enhancing interpretability and practical decision-making capabilities.

In summary, while shortest-path-based measures fail due to computational, structural, and interpretative issues, the proposed SCS provides a robust, interpretable, and efficient metric explicitly designed to capture spatial clustering patterns in GNN prediction errors. By clearly identifying spatially localized inaccuracies, SCS facilitates targeted interventions, model improvements, and robust deployments of GNN models in spatial applications.

## C  ADDITIONAL DETAILS ON EXPERIMENTS

In this appendix, we provide comprehensive details regarding the experimental setup, including datasets and tasks, baseline models, and training procedures, ensuring reproducibility and clarity.

## C.1 Testbed

Our experiments were conducted on a Dell PowerEdge C4140, The key specifications of this server, pertinent to our research, include:

**CPU:** Dual Intel Xeon Gold 6230 processors, each offering 20 cores and 40 threads.
**GPU:** Four NVIDIA Tesla V100 SXM2 units, each equipped with 32GB of memory, tailored for NV Link.
**Memory:** An aggregate of 256GB RAM, distributed across eight 32GB RDIMM modules.
**Storage:** Dual 1.92TB SSDs with a 6Gbps SATA interface.
**Networking:** Features dual 1Gbps NICs and a Mellanox ConnectX-5 EX Dual Port 40/100GbE QSFP28 Adapter with GPUDirect support.
**Operating System:** Ubuntu 18.04LTS.

We employed benchmark datasets that encompass node classification and node regression tasks to comprehensively assess our method across diverse spatial contexts. Specifically:

**Node Classification Datasets:**

- **Cora**(Sen et al., 2008): A citation network with 2,708 nodes representing scientific papers classified into 7 research categories. It has 5,429 citation links and sparse bag-of-words node features (dimension: 1,433).

- **Citeseer**(Sen et al., 2008): Another widely adopted citation network with 3,327 nodes and 4,732 edges. Papers are classified into 6 research categories, with node features represented by a 3,703-dimensional sparse bag-of-words.

**Node Regression Datasets:**

- **California Housing Prices**(Pace & Barry, 1997): A spatial regression dataset with 20,640 nodes representing geographic locations in California. The goal is to predict the median house prices based on spatial coordinates and associated features such as average income, population density, and proximity to various infrastructure.

- **Air Temperature Dataset**(Hooker et al., 2018): Contains temperature measurements from 1,305 meteorological stations globally, aiming to predict average air temperature based on geographical coordinates and associated climate factors.

**Synthetic Planar Datasets:** We synthesized two types of planar graphs to systematically analyze the impact of underlying graph structures on spatial error patterns:

- **Planar-Regular**: A uniformly connected planar graph generated with a regular node-degree distribution, consisting of nodes arranged in a grid-like spatial structure.

- **Planar-Power-Law**: A planar graph with power-law degree distribution (scale-free properties). The synthetic generation procedure followed Barabási–Albert preferential attachment (Barabási, 2013), modified to ensure planarity.

For all datasets, we either follow the default split (where provided) or followed standard data splits with ratios of 60% for training, 20% for validation, and 20% for testing. Results were averaged across five independent trials to ensure robustness and reproducibility.

## C.2 Baselines

We evaluated five representative baseline GNN architectures from three distinct methodological categories to comprehensively benchmark our proposed methods:

**Message-Passing GNNs:**

- **GCN**(Kipf & Welling, 2017): Graph Convolutional Network employing normalized adjacency-based feature aggregation.

- **GraphSAGE**(Hamilton et al., 2017): Aggregates features using neighborhood sampling and mean-pooling.

- **GAT** (Veličković et al., 2017): Graph Attention Network employing self-attention mechanisms to weigh neighboring nodes.

**Spectral-based GNN:**

- **ChebNet** (Defferrard et al., 2016): Spectral convolution network approximating filters via Chebyshev polynomial expansions of the graph Laplacian.

**Transformer-based GNN:**

- **Graph Transformer** (Dwivedi & Bresson, 2020): Leverages global self-attention mechanisms to capture long-range node dependencies without explicit reliance on local message-passing.

We utilized standard open-source implementations from widely adopted libraries (e.g., PyTorch Geometric, DGL) (Dwivedi et al., 2023; Hu et al., 2020), adhering strictly to published protocols and hyperparameter tuning recommendations for a fair comparison.

## C.3 TRAINING PROCEDURE

All models were trained following a rigorous, standardized training protocol to ensure fair and comparable evaluations across different methods and datasets:

- **Optimizer:** Adam with initial learning rates tuned in the range [0.001, 0.01]. The default best-performing value was typically 0.005 across datasets.
- **Weight Initialization:** Xavier initialization was uniformly applied to all models.
- **Epochs and Early Stopping:** Training was conducted for a maximum of 300 epochs, with early stopping activated based on validation performance to prevent overfitting (patience = 30 epochs).
- **Learning Rate Scheduler:** ReduceLROnPlateau scheduler was employed with a reduction factor of 0.5 and patience of 10 epochs.
- **Regularization:** Standard regularization techniques such as dropout (rates tuned from [0.1, 0.5]) and L2 weight decay (values tuned from $[10^{-4}, 10^{-3}]$) were employed across models.

For experiments involving our proposed SCA regularization, we trained each baseline model twice—once without SCA (standard baseline) and once with SCA integrated into the loss function. We tuned the regularization parameter $\lambda$ from the range [0.01, 1.0], ultimately selecting $\lambda = 0.1$ as the optimal value balancing structural error uniformity and predictive performance.

Hyperparameter optimization was performed using grid search on validation sets, and the final reported results are averages over five independently repeated runs with different random seeds, ensuring the statistical robustness of our conclusions.

## D FURTHER DISCUSSION

In this section, we provide a quantitative example to illustrate the limitations of existing GNN evaluation metrics, and we discuss how our proposed framework can be extended to capture spatial-temporal applications and other structural patterns.

## D.1 QUANTITATIVE EXAMPLE (TRAFFIC MANAGEMENT)

Consider two GNN models, Model I and Model II, tasked with predicting traffic flow in a city, measured in vehicles per hour (vph).

**Model I:**

- Errors occur at $k$ nodes, with a prediction error of 10 vph per node.

- The error distribution is uniformly spread across the network.

**Model II:**

- Errors occur at $k$ nodes, with a prediction error of 6 vph per node.
- The error distribution is spatially clustered at critical regions or intersections.

While traditional metrics like Mean Squared Error (MSE) would favor Model II due to its lower average error, our proposed Structural Cluster Statistic (SCS) metric, which quantifies the structural clustering of errors, would flag Model II as performing worse due to the concentration of errors in critical areas.

**Practical Implication**  Consider a critical intersection with a maximum capacity of 50 vph, fed by two adjacent roads, A and B, each expected to contribute 20 vph:

**Model I (uniform errors):**

- Makes a large error ($+10$ vph) only on Road A, while accurately predicting traffic flow on Road B.
- **Result:** The total flow remains 50 vph (i.e., $20 + 30 = 50$), avoiding congestion.

**Model II (clustered errors):**

- Makes smaller errors ($+6$ vph) simultaneously on both Roads A and B.
- **Result:** The total flow exceeds capacity (i.e., $26 + 26 = 52$ vph), causing congestion, despite the smaller individual errors.

This example illustrates how traditional metrics like MSE can favor models with lower average errors while overlooking critical operational risks. In contrast, our structure-aware SCS metric reveals the potential dangers of spatially clustered errors, offering a more nuanced evaluation.

D.2 FURTHER EXTENSION OF OUR FRAMEWORK

Our proposed evaluation framework, particularly the Spatial Cluster Statistic (SCS), is highly adaptable and can be extended to capture a wide range of spatial and structural error patterns beyond basic spatial clustering. In this section, we discuss several potential extensions of our framework, including how it can be adapted to detect errors related to spatial boundaries, directional biases, and temporal dependencies.

**Boundary Errors**  In many spatial tasks, errors near the boundaries of the graph may have a significantly different impact compared to errors in the interior. For example, in urban planning or environmental monitoring, boundary regions—such as city borders or edges of monitored areas—might have higher error tolerance or greater sensitivity. To address this, we can modify the weight assignments in our framework to give higher importance to errors occurring at boundary nodes, ensuring that our metric appropriately reflects the unique challenges of these critical areas.

**Directional Biases**  In certain applications, spatial directionality plays a key role in error distribution. For instance, in traffic flow predictions, errors may be more significant in certain directions (e.g., towards city centers during rush hour) compared to others. Our framework can be extended by assigning distinct weights based on the spatial direction of errors, allowing it to capture directional biases in prediction errors. This adaptation is particularly useful in modeling scenarios where spatial dependencies are not just local but also directional, such as weather patterns or traffic management.

**Temporal Dependencies**  Many real-world applications, such as traffic forecasting or environmental monitoring, involve dynamic systems where prediction errors evolve over time. To adapt our framework to such spatial-temporal settings, we can modify the weight assignment in Equation (3.1) to incorporate temporal dependencies. This adaptation would account for both spatial and temporal

proximity of errors, giving higher correlation weights to errors that occur closer in time. By incorporating temporal dependencies, our framework can capture how prediction errors evolve over time, providing more comprehensive evaluations in dynamic applications.

For example, in traffic prediction, errors at a given intersection may not only depend on the spatial proximity to other intersections but also on how traffic conditions change over time. This temporal extension makes our metric applicable to a wider range of applications involving dynamic systems.

**Generalizing to Other Structural Error Patterns**   Beyond spatial and temporal dependencies, our framework can be generalized to handle other structural error patterns by adjusting the weighting scheme. For instance, we can extend the metric to handle errors occurring along specific boundaries or errors that exhibit non-random patterns due to topological features of the graph, such as hub-based or scale-free structures. These extensions allow our framework to be applied in a wide range of contexts, including social networks, recommendation systems, and sensor networks, where the underlying graph structure significantly influences the error distribution.

In summary, our framework is highly flexible and can be extended to capture a variety of error patterns, including boundary effects, directional biases, and temporal dependencies. These extensions enhance the applicability of our evaluation metric in more complex, real-world scenarios, providing richer insights into model performance.

