# OpenReview forum: "Structural Error Patterns Matter: Towards More Structure-aware GNN Evaluation and Training"
_ICLR.cc/2026/Conference — ICLR 2026 Conference Withdrawn Submission_

### Official Review · Reviewer_ycLD · 2025-10-26

**Soundness:** 1
**Presentation:** 1
**Contribution:** 1
**Rating:** 0
**Confidence:** 5

**Summary:**

This paper suggests a new metric to compute errors in GNNs, called  Structural Cluster Statistic (SCS).

**Strengths:**

The idea to design a metric that can encapsulate how the miss-labeling of a GNN is spread across the graph is interesting and can be usefull.

**Weaknesses:**

1. The paper is hand-wavy and full of undefined terms. For example, the term "error pattern," which seems to be the main focus of this work, is not well-defined anywhere, neither in words nor mathematically. Figure 1 suggests that what the authors mean by "error pattern" is some continuous property over the graph topology of the structures that nodes that are mislabeled form. Nonetheless, even in "Figure 1", it is not mathematically defined what "random", "cluster", or "Dispersed Error" means. These terms must be rigorously well-defined. In the abstract, this is referred to as "structural error pattern"

2. The proposed method is not well-motivated. Lines 61 hand wave that "error patterns" are important, but they did not demonstrate in any example, mathematical formulation, nor references, the implication of not examining "error patterns", The fact that "error patterns" is not well defined as explained in (1), only makes it harder to understand what is the motivation for the proposed approach.

4. The evaluation section is weak for a paper with main contribution being an evaluation metric. I expect to see an exdtensive evaluation across many diverse datasets and GNNs. The authors uses 4 datasets, two of which are known to has many limitations including mislabeled examples [1].

5. It is not clear what contribution the proposed method provides. In the evaluation section, there is no demonstration of the insights that the new metric reveals. This seems natural to provide, especially as the author claims in their abstract, "A crucial but underexplored aspect of GNN evaluation is understanding how errors are distributed across the graph structure, which we refer to as the 'structural
error pattern”. If this is crucial, as the authors suggest, it would be beneficial to see at least one example of insights obtained by these patterns (assuming they are well-defined, which is not the case in this work)

6. The paper is not sound and poorly written. The claims raised in the abstract and across the paper, on the need for the proposed metric, are not demonstrated or shown anywhere to support this claim.

7. Not clear why the authors provided in Figure 1 as an example a grid graph. This emphasises the lack of motivation for the suggested approach as it was not even demonstrate on some real graph to show the need for the approach or what it can reveal over the graph.

[1] Position: Graph learning will lose relevance due to poor benchmarks, Bechler-Speicher et al., ICML 2025.

**Questions:**

1. What is "error pattern" ? please define this mathematically, and show how your metric provides insights into discovering this, mathematically.

2. Please provide examples of real use-cases where the defined "error pattern" above holds crucial information and show why existing metric do not capture that, on real examples.

3. Please well-define mathematically the 3 types of patterns in Figure 1, in a way that is consistent with them being "error pattern" (after this is also well-defined).

---

### Official Review · Reviewer_3ELp · 2025-10-30

**Soundness:** 3
**Presentation:** 2
**Contribution:** 2
**Rating:** 4
**Confidence:** 3

**Summary:**

This paper studies the structural error patterns in GNN models, proposes a new evaluation metric SCS explicitly to detect and quantify structural error patterns, and show that SCS can be adapted into a regularization framework during model training. The experiments results illustrate that the consideration of such factors can leading the errors distributed more uniformly.

**Strengths:**

1, It is interesting to study the error pattern in the GNN models.
2, The proposed metrics can reduce the chances of the error clusters in the GNN network.
3. Some applications might prefer a uniform distribution of the error loss.

**Weaknesses:**

1, The motivation of this paper should be more clear
2. The SCA regularization proposed do not always lead to improvement of overall performance
3. More competitors should be included in the experiments

**Questions:**

1.	The motivation in this paper should be clear, Currently, the paper has two objectives, the overall performance and error uniform distribution. When two factors conflict, what are your desired results. Even the overall performance is the same, is the uniform error distribution better than other case? In Figure 1, is it meaningful to just distribute errors with the same performance? Suppose each node has the same weight, the overall performance is more important metric than the error distribution.

2.	The value of the error distributions study should improve the overall performance. However, we can see that from Figure 1, the overall performance is not improved, and sometime degraded, which damages the value of the proposed methods.

3.	It might result in more challenges to improve the overall performance of GNN, if we apply SCA regularization to achieve structural uniformity of errors, as it is hard to capture the error pattern. For example, for the cluster error in Figure 1b, it is relatively easy to detect the error, which then further provides chances to fix them.\

4.	The compared methods are not new. GCN, GAT are very early works. Some work on heterophily graph might be related to your work, and can be compared in the experimental study

5.	This paper applies SCA in the GNN training. It is better to discuss such a regulation works on graphs with different distributions, such as power-law graph.

6.	The paper mainly studies the structural factors related to error patterns. How about other factors like content/label of neighbor nodes.

---

### Official Review · Reviewer_YGt3 · 2025-10-31

**Soundness:** 1
**Presentation:** 2
**Contribution:** 1
**Rating:** 2
**Confidence:** 5

**Summary:**

The paper argues that standard GNN metrics (ACC/MSE) are exchangeable and therefore blind to the spatial distribution of errors (e.g. how errors are arranged on the graph). The authors thus propose a graph-aware diagnostic, called the Structural Cluster Statistic (SCS), which is essentially Moran's $I$ over residuals on the test subgraph, and a regularizer, SCA, given by a squared version of SCS added to the loss to discourage residuals from being clustered. Experiments show SCS reveals clustered errors and that SCA reduces SCS at a small cost in ACC/MSE.

**Strengths:**

1) Some motivation for structure-aware diagnostics; Fig.\1 illustrates a failure mode of exchangeable metrics;
(2) The solution is a simple, computable ($O(|E|)$) statistic tied to a well-known spatial measure (Moran's $I$).

**Weaknesses:**

Overall, while I do appreciate the importance of diagnostics that would reflect patterns of correlations between nodes in the graph, I do not necessarily understand the authors’ proposed solution. Here are my main reservations and questions.

1) **Theory is not convincing as stated.** Theorem 3.1 claims that if two error distributions have the same expected sum $\mathbb{E}[S(\epsilon)]$, then the expectation of any exchangeable metric should agree under both. This seems a little odd. Consider a simple setting, with two errors $\epsilon_1, \epsilon_2$. Suppose that under distribution P, $\epsilon_1 ,\epsilon_2 ~$ Uniform(0,1), so that the $\epsilon_1 +\epsilon_2 \sim Uniform(0,2)$. Under distribution 2, $\epsilon_1 \sim Bernouilli(0.5), \epsilon_2= \epsilon_1$. The expectations of the sums are equivalent (all equal to 1).  But the expectation of the maximum (which is exchangeable) is 2/3 in the first case, and 0.5 in the second. Wouldn't this contradict the theorem?

2) **The motivation is too unclear.** While I agree that patterns of autocorrelation in residual structures could be better investigated in GNNs, the authors' argument for its necessity is not rigorous. They claim that "Without structure-aware evaluation, practitioners lack insights into how prediction errors manifest across the graph, hindering their ability to diagnose, address, and prevent localized failures effectively," and that diagnosing errors is necessary to improve the fit. However, the problem is not well-formulated mathematically. From the authors' comments in the paper, it is unclear if the problem is to find local failure modes (e,g, sensor malfunctions), or to allow to select models. The problem could be formulated more mathematically. For instance, consider a regression task, with $f$ as a GNN function. The standard regression model assumes: $f(X) = \mathbb{E}[Y|X]$, so that $y = f(x) + \epsilon$, where $\epsilon$ indicates iid noise. Residuals in the errors could indicate two things:
       1)  *Residual structure in $\epsilon$ (model mis-specification)*: In this case, $\epsilon$ is not iid and depends on $x$/ the graph, likely necessitating a different architecture (e.g. more complex, to explain more of the signal).
        2)  *Autocorrelation in $\epsilon$:* The structure is well-approximated, but $\epsilon$ is not iid, yet independent of $X$. For example, in traffic prediction, heavy rainfall might correlate traffic deviations in a neighborhood. In this case, autocorrelation is not a problem for the fit, but it would be at inference time when estimating variance, etc. For instance, if the SCS metric shows correlation in the error, then it could be argued that randomly assigning nodes to train/validation/test splits will probably induce data leakage.


3) **I find part 2  (SCA) confusing and potentially detrimental to the paper.** Why is regularizing towards a solution that "looks good" beneficial? The first part of the paper focuses on diagnostics, while the second seems to suggest training the model to perform well on the test set. However, regularizing a model away doesn't address the underlying issues. If the goal is to diagnose a lack of fit in the GNN class, how does this regularization strategy that would essentially mask it contribute to that?  An alternative could have been to use SCS for model selection: i.e. amongst models with different architectures, use SCS to choose which architecture provides the better fit.


4) Moran's I is usually considered to be test statistic --- not a metric---, and not particularly interpretable as such. It is usually standard practice to use it to test the null (e.g no spatial clustering). How do the authors evaluate whether their SCS is within expected variation?

5) **Lack of comparisons to other metric**: finally, the authors could have argued that the suite of tools developed for spatial statistics could be pertinent, --- or at least mention these techniques. Examples include local Moran's I, Geary's C --- or measures of patchiness (e..g CHAOS https://www.nature.com/articles/s41467-022-34879-1#Sec10)

**Questions:**

1) Is the test set assumed to be a fully connected graph? How do the authors ensure that the statistic can be computed? (e.g. if the test set is 20\% of the data, chosen randomly in the graph, wouldn't we expect the graph induced by the test nodes to be pretty sparse and disconnected)?

---

### Official Review · Reviewer_ozky · 2025-10-31

**Soundness:** 2
**Presentation:** 2
**Contribution:** 2
**Rating:** 2
**Confidence:** 4

**Summary:**

This paper argues that commonly used GNN evaluation metrics such as accuracy  are exchangeable, and cannot distinguish different structural error patterns across the graph. Based on the limitation, the authors propose a new metric SCS and further introduces SCA regularization, which incorporates a squared form of SCS into the training loss to discourage spatially clustered errors.

**Strengths:**

The paper identifies a relevant and underexplored issue in GNN evaluation: traditional metrics do not capture spatial or structural patterns in prediction errors.

The theoretical analysis clarifies why exchangeable metrics cannot differentiate different error distributions across the graph, which is conceptually insightful.

The idea that where errors occur matters, not just how many, is intuitive and aligns with practical concerns in spatial/graph applications.

**Weaknesses:**

1. SCS may be confounded by graph structure rather than model behavior.  For instance, in Figure 2 (planar power-law case), SCS primarily reflects the underlying graph topology (e.g., the presence of hubs) rather than revealing model-specific failure patterns. This challenges the core claim that SCS robustly detects meaningful structural clustering of errors.

2. The role of SCA is unclear when error clustering is induced by graph structure. If the clustering arises because of the inherent graph structure rather than model shortcomings, then penalizing it with SCA may not be meaningful. The paper does not clarify when SCA is beneficial versus when it may suppress necessary model behavior.

3. Experimental results do not convincingly demonstrate benefit.
- In Table 1, adding SCA often reduces accuracy or increases MSE, suggesting the method may impair predictive quality.
- It is unclear whether lower SCS is inherently desirable, especially when accompanied by lower predictive performance.
- The experiments do not justify the trade-off or provide scenarios where the trade-off is beneficial.

4. Missing comparisons to alternative structural metrics / baselines.
 To claim the necessity or superiority of SCS, comparisons to other graph-based or spatial autocorrelation measures (or even naive distance/clustering baselines) are needed.

5. Experiment scope is limited.
 Only four datasets and five backbones are tested; no ablation study, no real-world scenario, and no evaluation where structural clustering actually matters for decision-making.

**Questions:**

N/A

---

### Note · Authors · 2026-01-08

I have read and agree with the venue's withdrawal policy on behalf of myself and my co-authors.